# Inhibition of Ribosome Assembly and Ribosome Translation Has Distinctly Different Effects on Abundance and Paralogue Composition of Ribosomal Protein mRNAs in *Saccharomyces cerevisiae*

Md Shamsuzzaman,[a]* Nusrat Rahman,[a]§ Brian Gregory,[a] Ananth Bommakanti,[a]◇ Janice M. Zengel,[a] Vincent M. Bruno,[b] Lasse Lindahl[a]

aDepartment of Biological Sciences, University of Maryland—Baltimore County, Baltimore, Maryland, USA
bInstitute for Genome Sciences, Baltimore, Maryland, USA

**ABSTRACT** Many mutations in genes for ribosomal proteins (r-proteins) and assembly factors cause cell stress and altered cell fate, resulting in congenital diseases collectively called ribosomopathies. Even though all such mutations depress the cell's protein synthesis capacity, they generate many different phenotypes, suggesting that the diseases are not due simply to insufficient protein synthesis capacity. To learn more, we investigated how the global transcriptome in *Saccharomyces cerevisiae* responds to reduced protein synthesis generated in two different ways: abolishing the assembly of new ribosomes and inhibiting ribosomal function. Our results showed that the mechanism by which protein synthesis is obstructed affects the ribosomal protein transcriptome differentially: ribosomal protein mRNA abundance increases during the abolition of ribosome formation but decreases during the inhibition of ribosome function. Interestingly, the ratio between mRNAs from some, but not all, pairs of paralogous ribosomal protein genes encoding slightly different versions of a given r-protein changed differently during the two types of stress, suggesting that expression of specific ribosomal protein paralogous mRNAs may contribute to the stress response. Unexpectedly, the abundance of transcripts for ribosome assembly factors and translation factors remained relatively unaffected by the stresses. On the other hand, the state of the translation apparatus did affect cell physiology: mRNA levels for some other proteins not directly related to the translation apparatus also changed differentially, though not coordinately with the r-protein genes, in response to the stresses.

**IMPORTANCE** Mutations in genes for ribosomal proteins or assembly factors cause a variety of diseases called ribosomopathies. These diseases are typically ascribed to a reduction in the cell's capacity for protein synthesis. Paradoxically, ribosomal mutations result in a wide variety of disease phenotypes, even though they all reduce protein synthesis. Here, we show that the transcriptome changes differently depending on how the protein synthesis capacity is reduced. Most strikingly, inhibiting ribosome formation and ribosome function had opposite effects on the abundance of mRNA for ribosomal proteins, while genes for ribosome translation and assembly factors showed no systematic responses. Thus, the process by which the protein synthesis capacity is reduced contributes decisively to global mRNA composition. This emphasis on process is a new concept in understanding ribosomopathies and other stress responses.

**KEYWORDS** cell stress, nucleolar stress, ribosomal protein paralogues, ribosomal proteins, ribosome biogenesis, transcriptome, translation, translation stress

Address correspondence to Lasse Lindahl, lindahl@umbc.edu.

*Present address: Md Shamsuzzaman, Bristol Myers Squibb, Lawrenceville, New Jersey, USA.

§Present address: Nusrat Rahman, American Psychiatric Association, Washington, DC, USA.

◇Present address: Ananth Bommakanti, Twist Biosciences Singapore, Singapore.

The authors declare no conflict of interest.

Eukaryotic ribosome formation involves synthesizing and assembling four rRNA molecules with 79 to 80 unique ribosomal proteins (r-proteins). More than 250 proteins and noncoding (ncRNA) molecules are required for this complex process (1–4). The number of ribosomes per cell ($\sim10^5$ in yeast, $\sim10^7$ in humans) and their structural complexity make ribosome formation very resource-intensive. Mechanisms have evolved to match ribosome biogenesis to growth conditions and growth rate (5–7). For example, the expression of rRNA and r-protein genes is repressed during nutritional insufficiency, while genes encoding proteins needed to adapt to the stress condition are upregulated (8, 9). The nutrient shortage inactivates TORC1, a major gatekeeper of ribosome gene transcription, resulting in the replacement of positive transcription factors on ribosomal genes with negative ones. The key role of ribosome metabolism in general cell physiology is also illustrated by the signals produced during the distortion of normal ribosome function which may result in apoptosis and cell death (10–13).

Mutations in numerous genes for r-proteins or assembly factors cause a variety of congenital human diseases, collectively called ribosomopathies (14–17). It is assumed that the root for these calamities is the decreased translation capacity, which in turn can lead to the formation of specialized ribosomes with altered preferences for specific mRNAs, the interaction of free r-proteins with p53 and cell cycle regulators, and changes to specific metabolic pathways (18–23). One of the standing puzzles of ribosomopathies is that, although mutations in different ribosomal genes all decrease the rate of ribosome formation, they generate a variety of distinct disease phenotypes and, vice versa, mutations in different r-protein genes can cause the same disease, e.g., Diamond Blackfan anemia (24).

To shed light on the effects of different assaults on the cell's capacity for protein synthesis, we used *Saccharomyces cerevisiae* (yeast) to compare the global transcriptomes after reducing the protein synthesis capacity by (i) blocking ribosome formation by abrogating the synthesis of r-protein ribosome uL4 (nucleolar stress, also called ribosome stress), and (ii) abolishing ribosome function by stopping the synthesis of translation elongation factor 3 (eEF3; translation stress). Both acts lowered the translation capacity but in completely different ways. Blocking uL4 synthesis reduced the accumulation of newly synthesized ribosomes without affecting the ability of preexisting ribosomes to translate mRNA. Conversely, abrogating eEF3 synthesis did not affect ribosome number but progressively inhibited ribosome function (Fig. 1). Our results showed that the two stresses have profoundly different effects on the transcriptome.

## RESULTS

**Establishing nucleolar and translation stresses.** To establish nucleolar and translation stresses, we used strains in which the only gene for r-protein uL4 or translation elongation factor eEF3 is controlled by the *GAL1/10* promoter inserted at the normal genomic position of the genes *RPL4A* or *YEF3/TEF3* (see Table S1). We refer to these strains as Pgal-uL4 and Pgal-eEF3. In galactose medium, the *GAL1/10* promoter was active, and both strains grew with a doubling time of $\sim2.1$ h at 30°C. Sucrose gradient analysis indicated that ribosome assembly and translation were normal: the distribution of free ribosomal subunits, 80S, and polysomes did not differ from wild type, and no abnormal ribosome peaks were seen (Fig. S1A, C, and F) (25, 26).

Shifting Pgal-uL4 from galactose to glucose medium repressed the *GAL1/10* promoter and abolished the synthesis of uL4, blocking ribosome assembly and slowing growth because the assembly of new ribosomes was obstructed (Fig. 2A and Fig. S1D). This decrease in the number of ribosomes was independent of whether *RPL4A* (used in all experiments except that shown in Fig. S1E) or its paralogue, *RPL4B*, was the source of uL4 protein (compare Fig. S1D and E). In all experiments reported here, except for that shown in Fig. S1E, *RPL4B* was inactivated and *RPL4A* was the only source of r-protein uL4. Moreover, abolishing the synthesis of other r-proteins generated the same pattern, suggesting that the consequences of the abolition of uL4 synthesis were representative of nucleolar stress in general (25).

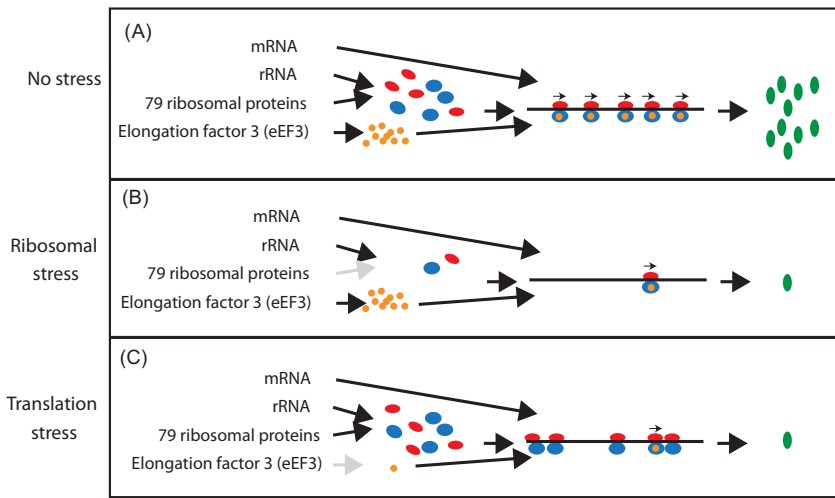

**FIG 1** Stress conditions used for RNA-seq analysis. (A) Exponential growth with uninhibited protein synthesis. The 40S and 60S ribosomal subunits are shown in red and blue, respectively, eEF3 is in orange, and protein products are in green. Arrows above the 80S ribosomes on the mRNA indicate elongation-competent ribosomes, each associated with an eEF3 molecule. (B) Nucleolar stress. Ribosome assembly is inhibited by repression of uL4 synthesis, resulting in reduced numbers of ribosomes relative to cell mass, but the remaining ribosomes are peptide elongation-competent because the number of eEF3 molecules is not decreased. (C) Translation stress. The number of ribosomes is unchanged, but ribosomal translocation on the mRNA is inhibited as eEF3 is depleted. See Fig. S1 and references 25, 26, and 56 for further details.

Shifting Pgal-eEF3 to glucose medium decreased the number of eEF3 molecules (26), which slowed growth (Fig. 2B), since eEF3 is essential for ribosome translocation but does not affect the cell concentration of ribosomes (Fig. S1F and G). Despite the decreasing synthesis of eEF3, the growth rate initially increased, the result of an upsurge in the differential

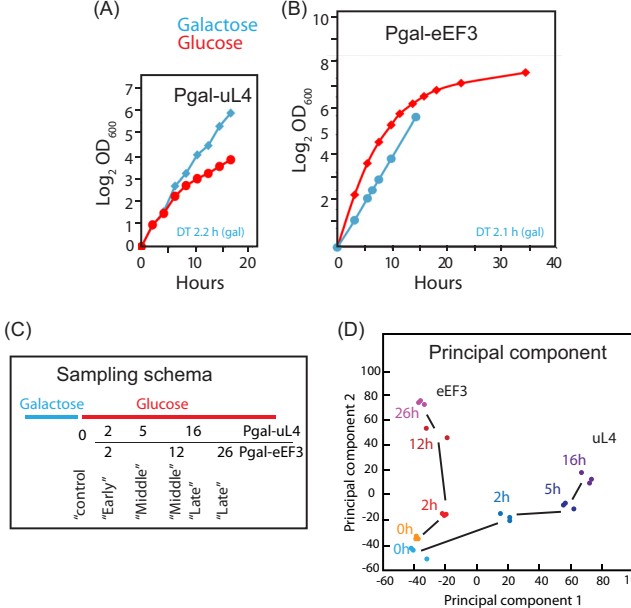

**FIG 2** Repression of uL4 and eEF3 synthesis. (A and B) Growth curves of Pgal-uL4 and Pgal-eEF3 growing in galactose (blue) and after a shift from galactose to glucose medium (red). (C) Schema for the sampling of cultures. The sampling times were selected to ensure that the growth rates ($\Delta OD_{600}/\Delta t$) of the Pgal-uL4 and Pgal-eEF3 cultures were approximately equal at the time of withdrawing the control, early, middle, or late samples. The sampling times of each sample are indicated. (D) Principal-component analysis of the RNA-seq data in samples taken at the indicated times in the two strains. Results from Pgal-uL4 are shown in blue colors and Pgal-eEF3 results are shown in yellow to red colors.

**TABLE 1** Differentially expressed genes after repressing synthesis of r-protein uL4 or translation of (eEF3)[a]

| | Total DEGs | | | | | | | | Up-DEGs | | | | | | | | Down-DEGs | | | | | | | |
|---|---|---|---|---|---|---|---|---|---|---|---|---|---|---|---|---|---|---|---|---|---|---|---|---|
| | Pgal-uL4 | | Both | | Pgal-eEF3 | | All | | Pgal-uL4 | | Both | | Pgal-eEF3 | | All | | Pgal-uL4 | | Both | | Pgal-eEF3 | | All | |
| Time | n | F | n | F | n | F | n | F | n | F | n | F | n | F | n | F | n | F | n | F | n | F | n | F |
| Early | 334 | 6.2% | 223 | 4.2% | 104 | 1.9% | 661 | 12.4% | 128 | 2.4% | 17 | 0.3% | 22 | 0.4% | 167 | 3.1% | 206 | 3.9% | 206 | 3.9% | 82 | 1.5% | 494 | 9.2% |
| Middle | 535 | 10.0% | 307 | 5.7% | 296 | 5.5% | 1138 | 21.3% | 334 | 6.2% | 41 | 0.8% | 89 | 1.7% | 464 | 8.7% | 201 | 3.8% | 266 | 5.0% | 207 | 3.9% | 674 | 12.6% |
| Late | 703 | 13.1% | 296 | 5.5% | 410 | 7.7% | 1409 | 26.3% | 447 | 8.4% | 76 | 1.4% | 175 | 3.3% | 698 | 13.1% | 256 | 4.8% | 220 | 4.1% | 235 | 4.4% | 711 | 13.3% |
| All | 165 | 3.1% | 116 | 2.2% | 55 | 1.0% | 336 | 6.3% | 4 | 0.1% | 0 | 0.0% | 12 | 0.2% | 16 | 0.3% | 161 | 3.0% | 116 | 2.2% | 43 | 0.8% | 320 | 6.0% |

[a]n, number of DEGs in the sample; F, percentage of DEGs normalized to the total number of genes.

rate of ribosome formation after the nutritional shift up (Fig. 2B) (6, 27). As the number of eEF3 molecules decreased, the growth rate progressively decreased. No shift-up effect was seen in the growth curve for Pgal-uL4, concurring with the abolition of ribosome production after the shift to glucose medium, which prevented an increase in ribosome number (Fig. 2A). Previous results indicated that the effect of the shift up on gene expression stabilized within an hour after the shift (6); we chose 2 h post-shift-up as the time to begin analyzing the transcriptome in cells subjected to either of the two assaults on the protein synthesis capacity (Fig. 2C).

**General trends of the transcriptome.** We collected RNA samples from triplicate independent cultures of Pgal-uL4 and Pgal-eEF3 before and at the indicated times after shifting from galactose to glucose medium. The sampling times were chosen to ensure approximately equal growth rates of the two cultures at the time of sampling, as estimated from the slopes of the respective growth curves [the $(\Delta OD/\Delta t)_{\text{sampling time}}$]. We refer to the samples as control, early, mid, and late (Fig. 2C). Bio-Analyzer electrophoretograms showed high rRNA integrity in all but one of the middle samples from Pgal-eEF3, which was therefore discarded. The remaining 23 samples were subjected to RNA sequencing (RNA-seq) analysis; mapping the results to the *S. cerevisiae* genome (https://www.yeastgenome.org/) yielded $2.3 \times 10^7$ to $3.3 \times 10^7$ reads from each sample (Data Set 1A). The abundance of mRNA from specific genes (expression level) in each sample was calculated by mapping reads to each gene or open reading frame in the yeast genome (gene read counts) (Data Set 1B), and the change in the expression of each gene (fold change) after the onset of stress was calculated as the read counts in each sample normalized to the read counts in the control sample of the same strain. Data Set 2A lists the $\log_2$ fold change (lfc) for 5,348 open reading frames, all of which had at least 50 read counts at all sampling times.

To ascertain the similarity between biological replicates from a given strain and the variance among depletion samples collected from Pgal-uL4 and Pgal-eEF3, we performed a principal-component analysis (PCA) on the read counts per transcript or gene feature after size factor normalization using the DESeq2 package. The replicate samples for a given time point in each strain clustered tightly, indicating high reproducibility (Fig. 2D), as was also indicated by the *P* values and false-discovery rates (FDR) for genes whose expression increased up or down by 2-fold or more ($|\text{lfc}| \geq 1$) (Data Set 2A).

After the induction of stress, the results moved along different paths in the PCA space (Fig. 2D), indicating that the transcriptome pattern developed differently during the two stress forms. This was in concordance with the distribution of differentially expressed genes (DEGs, with at least a 2-fold change in mRNA abundance up [up-DEGs] or down [down-DEGs]) and an FDA of <0.05. DEGs represented 12 to 26% of the total number of genes at different times, but only 21 to 35% of all DEGs were observed in both strains at a given time. Furthermore, only 6% of genes always had an $|\text{lfc}|$ of $\geq 1$ (Table 1). Specific DEGs at each time point are listed in Data Set 3A.

**Characteristics of the transcriptome after the shift to glucose medium.** The ratios of the read counts in the control samples ($t = 0$) were between 0.70 and 1.3 for 79% of the 5,348 genes listed in Data Set 1. Thus, the baseline expression of most genes was similar in the two strains before the onset of stress. Shifting Pgal-uL4 from galactose to glucose medium reduced the pool of uL4A mRNA by 30- to 60-fold and

elevated the eEF3 mRNA abundance by <50% (Data Set 2A). In contrast, shifting Pgal-eEF3 to glucose medium reduced the pool of eEF3 mRNA by 30- to 100-fold, while the abundance of uL4 mRNA was unchanged. Thus, the repression of uL4 and eEF3 mRNA was specific to each of the two strains.

Besides the exposure to either nucleolar or translation stress, mRNA abundance might also be affected by a change of carbon source. To parse the consequences of these different interferences, we first identified genes whose expression changed most differently, or most similarly, by using a rank value difference (RVD) analysis (28). This involved ranking the lfc values for each gene within each sample, calculating the RVD for the corresponding genes in early, middle, and late samples from the two strains, and finally ranking the sum of the absolute RVDs for each gene at the three time points ($\sum$|RVD|) (Data Set 2B). The genes with the highest $\sum$|RVD| were most likely to show different expression patterns, on average, in the two strains after the induction of stress. Conversely, on average, the genes with the lowest $\sum$|RVD| were most likely to have similar expression patterns.

**(i) Many mitochondria-related genes were repressed during both nucleolar and translation stress.** Submitting the 200 genes with the lowest $\sum$|RVD| to the GO enrichment analysis tool of the Gene Ontology database (http://geneontology.org/) returned several GO terms for genes mostly encoding mitochondrial proteins (Fig. 3). The average ratio between the read counts for these genes in the Pgal-uL4 and Pgal-eEF3 control galactose cultures was 1.2 (Fig. 3, column 1), indicating that their expression levels were similar in the two strains before the imposition of stress. After the switch to glucose medium, most of the genes were repressed in both strains (Fig. 3, columns 2 to 7). Moreover, mRNAs from the low-affinity glucose transporter of the major facilitator superfamily, *HXT1* and *HTX3*, were induced. We concluded that the pattern of expression of genes with the lowest $\sum$|RVD| suggested that they were under glucose repression, in accordance with the previous analysis of the regulation of genes for the formation of mitochondria (29). Note that the lfc for the mitochondrial genes changed little after 2 h, indicating that the glucose repression was established and stable within the first 2 h after the medium shift (Fig. 3, columns 8 and 9).

**(ii) The abundance of mRNA for r-proteins changed differently during nucleolar and translation stress.** Submitting the 200 genes with the highest $\sum$|RVD| to the GO term enrichment yielded a single GO term, cytoplasmic translation, which was enriched by 3.4-fold (FDR, 2.2E−02). This prompted us to generate heatmaps of all genes encoding r-proteins, ribosome assembly (Ribi) factors, and ribosome translation factors (Fig. 4 and 5). The mRNAs for these genes accounted for about 20%, 3%, and 3% of all mRNAs, respectively, in the galactose control cultures (*t* = 0) for both strains. Furthermore, the average ratios of the read counts for Pgal-uL4 and Pgal-eEF3 before the imposition of stress were between 0.8 and 1.2, indicating similar baseline expression in the control cultures (column 1 of Fig. 4 and 5; Fig. S3).

Interestingly, r-protein mRNA abundance was regulated in opposite directions during nucleolar and translation stress. During the repression of uL4 synthesis, mRNAs for other r-proteins were initially repressed, then switched to induction after 2 h (Fig. 4, columns 2 to 4). Repression of eEF3 synthesis resulted in the opposite pattern: an initial induction as expected from the nutritional shift up followed by repression (Fig. 4, columns 5 to 7). Importantly, as opposed to the genes for mitochondrial proteins, the lfc for r-protein genes kept changing after 2 h, indicating that the r-protein genes were regulated by the imposed stress rather than the medium shift (see columns 8 and 9 in Fig. 4).

The lfc values for some samples had FDR values of >0.05, since they were close to 0. Consequently, we plotted all r-protein mRNA lfc values against time, which showed that essentially all lfc values fell on curves with positive slopes after inhibiting uL4 synthesis and negative slopes after repressing eEF3 synthesis (Fig. S2). The only exceptions were the values for RPS22B, which declined after abrogating uL4 synthesis. Overall, we concluded that the mRNA for r-protein genes was induced during nucleolar stress but repressed during translation stress.

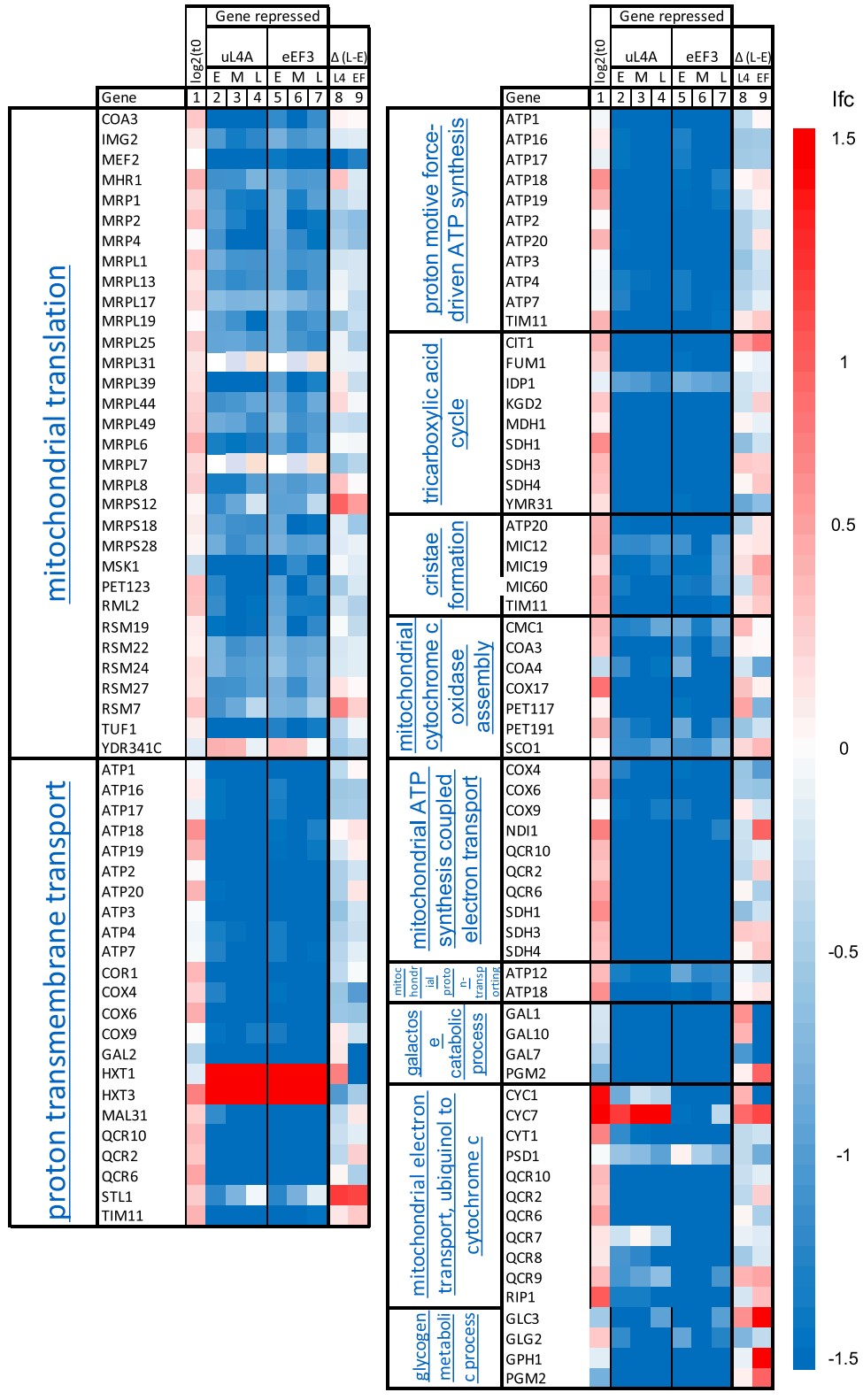

**FIG 3** Heatmaps for genes with the most similar change of expression (lfc) after the shift from galactose to glucose medium. Submission of the 200 genes with the lowest rank value difference (Data Set 2B; see also text) to GO enrichment analysis at http://geneontology.org/ returned the biological process GO terms (with FDRs in parentheses): mitochondrial translation (4E−12), proton transmembrane transport (1E−3), proton force-driven ATP synthesis (4E−3), tricarboxylic cycle (4E−2), cristae formation (4E−3), mitochondrial cytochrome *c* oxidase assembly (1E−2), mitochondrial proton-transporting ATP synthesis complex assembly (2E−2), galactose

**(iii) Relative expression of paralogues for some r-protein genes changed during stress.** About two-thirds of the *S. cerevisiae* r-proteins are encoded by pairs of paralogous genes, called A and B, whose protein products may differ by a few amino acids. To determine if the ratio of paralogous messengers changed during stress, we calculated the ratios of read counts for the A and B alleles normalized to the corresponding ratios in the control cultures. Note that we could not determine the absolute ratio between the expression of the paralogous allele, as we did not know if the RNA-seq procedure registered both alleles with the same efficiency, but this uncertainty was eliminated by normalizing the data to the control sample ($t = 0$). The results showed that the relative expression of paralogue pairs changed more than 2.5-fold for uS8, eS28, uL6, eL18, eL22, uL30, and eL33 but less than 50% for most other r-protein paralogue pairs (Fig. 6).

**(iv) Genes for ribosome assembly and ribosome translation did not change coordinately with r-protein genes or with each other.** Ribosome homeostasis and function depend on hundreds of auxiliary factors that support ribosome formation and mRNA translation. Unexpectedly, the mRNA abundance for neither the ribosome assembly factors (Ribi) nor the translation factors was coordinated with the changes in the r-protein mRNAs (Fig. 5). The lfc for most of these mRNAs changed less than 2-fold and showed no obvious common pattern. The mRNAs for nuclear pore proteins were also not coordinated with r-protein mRNA, even though about 1,000 precursor ribosomes are exported through the nuclear pores from the nucleus to the cytoplasm each minute during exponential growth (30) (Fig. 7).

**(v) Other GO terms.** A search for other patterns among DEGs showed that GO terms defined by DEGs at specific time points in each strain mostly included genes for metabolic functions, while others were closely related to GO terms for glucose-repressed genes mentioned above (Data Set 3).

The analysis based on rank value differences specifically identified genes with the same or different regulation patterns in response to the two types of stress. To expand our scrutiny for genes whose expression changed significantly during the stresses, we also searched for genes with a |Δlfc| of ≥1. Subjecting the resulting gene lists to GO term enrichment analysis identified GO terms for genes in protein transport and the mitotic cell cycle, which were not identified in the RVD analysis (Data Set 4).

## DISCUSSION

Interference with the translation machinery's formation or function often has serious consequences for an organism's fitness, resulting in various human diseases. It is assumed that insufficient translation capacity is a significant contributor to a disease state, but it is not clear whether reduced translation capacity as such, no matter the cause, always has the same effect on the function of the cell. To answer this question, we compared the yeast transcriptome after abolishing ribosome biogenesis (nucleolar stress) and blocking ribosome translocation (translation stress). Both stresses inhibited protein synthesis, cell growth, and cell cycle progression, but it was unknown if the transcriptomes associated with these calamities were the same or different. Our results showed that both the r-protein mRNA abundance and paralogue composition of mRNAs were regulated differently during the two stresses.

**FIG 3** Legend (Continued)

metabolic process (2E−2), mitochondrial metabolic process, ubiquinol to cytochrome *c* (4E−2), and glycogen metabolic process (5E−2). The heatmaps show lfc for genes listed collectively under these GO terms. The first two columns show GO terms and the gene names (https://www.yeastgenome.org/), respectively. Column 1 shows $\log_2$ (read count in Pgal-uL4/read count in Pgal-eEF3) at $t = 0$. Columns 2 to 4 show the lfc ($\log_2$ of fold change) in Pgal-uL4 gene expression (read count for the early [E], middle [M], and late [L] samples, respectively, relative to Pgal-uL4 read count at $t = 0$ [control]). Columns 5 to 7 similarly show the lfc in Pgal-eEF3 gene expression in the E, M, and L samples relative to the control culture. Columns 8 and 9 show the differences between lfc in the L and E samples for Pgal-uL4 and Pgal-eEF3, respectively, i.e., the changes in gene expression between the late and early time points in each strain. The scale for the heatmaps is shown to the right. The heatmap scale from −1.5 to +1.5 is shown to the right.

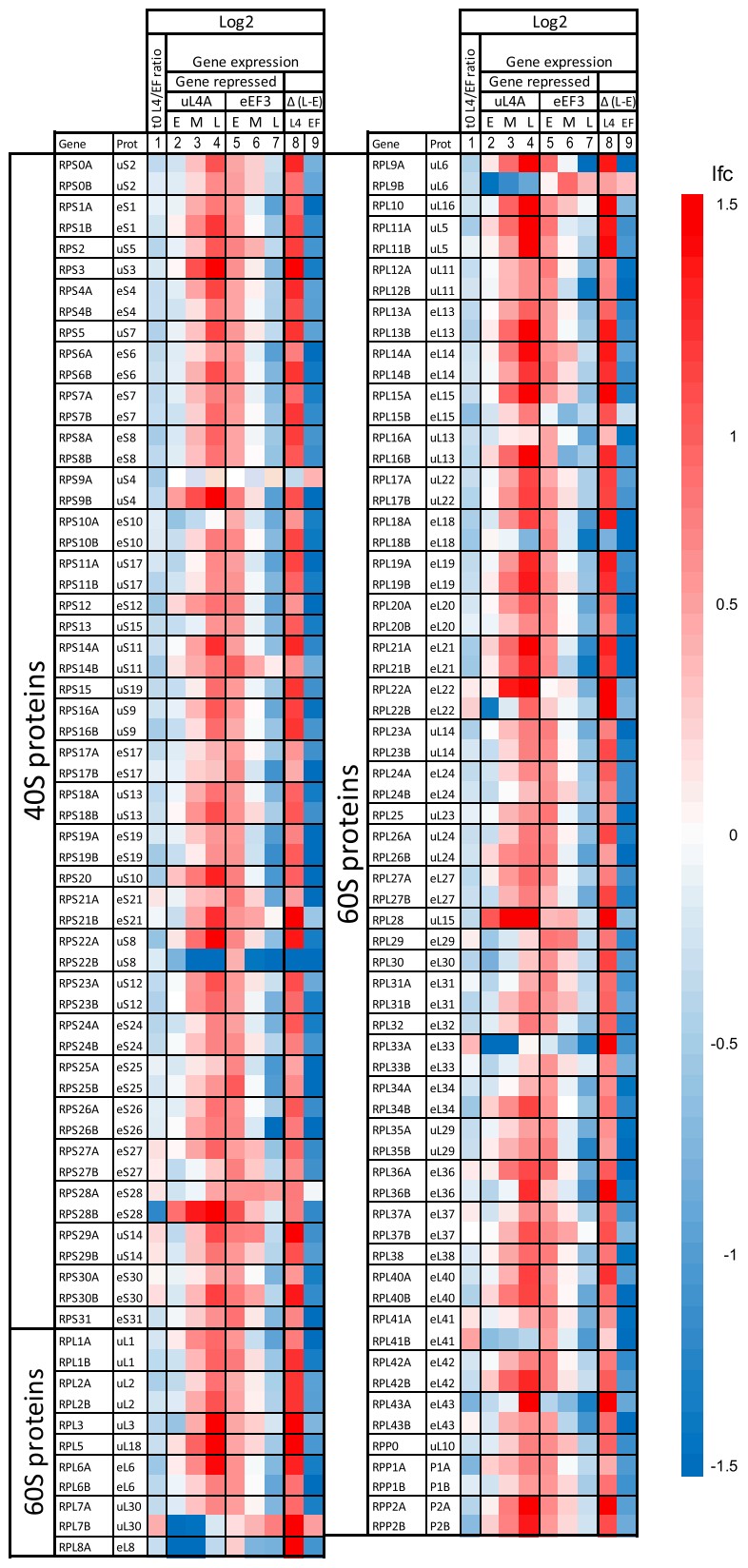

**FIG 4** Heatmaps for genes for the 40S and 60S cytoplasmic ribosomal proteins during nucleolar and translation stress. The two left columns show gene names (https://www.yeastgenome.org/) and the names of their encoded proteins according to the nomenclature of Ban et al. (57). The heatmaps are arranged as explained in the legend in Fig. 3.

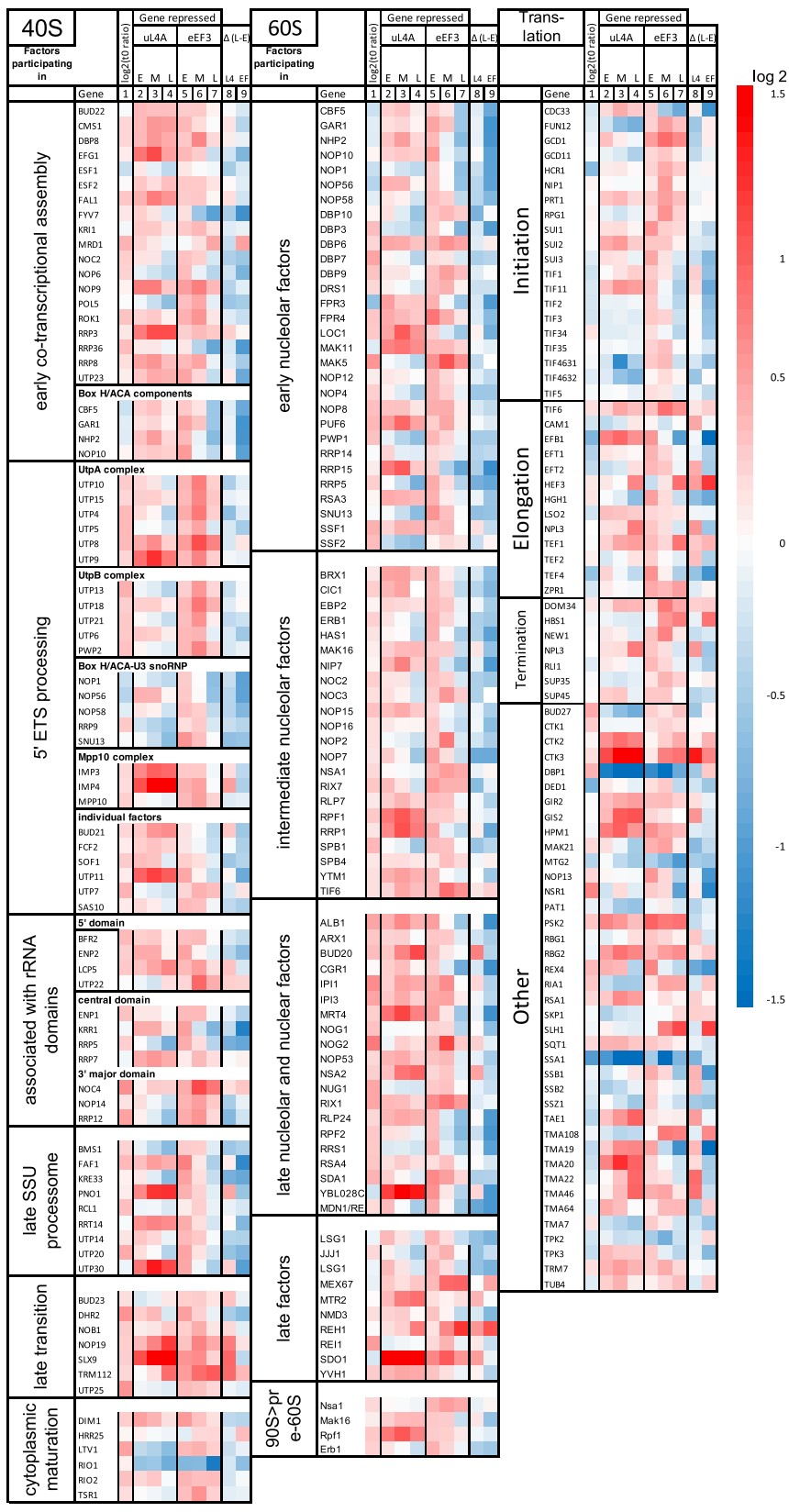

**FIG 5** Heatmaps for the 40S and 60S assembly factors and translation factors during nucleolar and translation stress. The columns are organized as explained in the legend to Fig. 3. The genes are ordered according to their function in the assembly of each subunit or ribosome translation function (4, 58, 59).

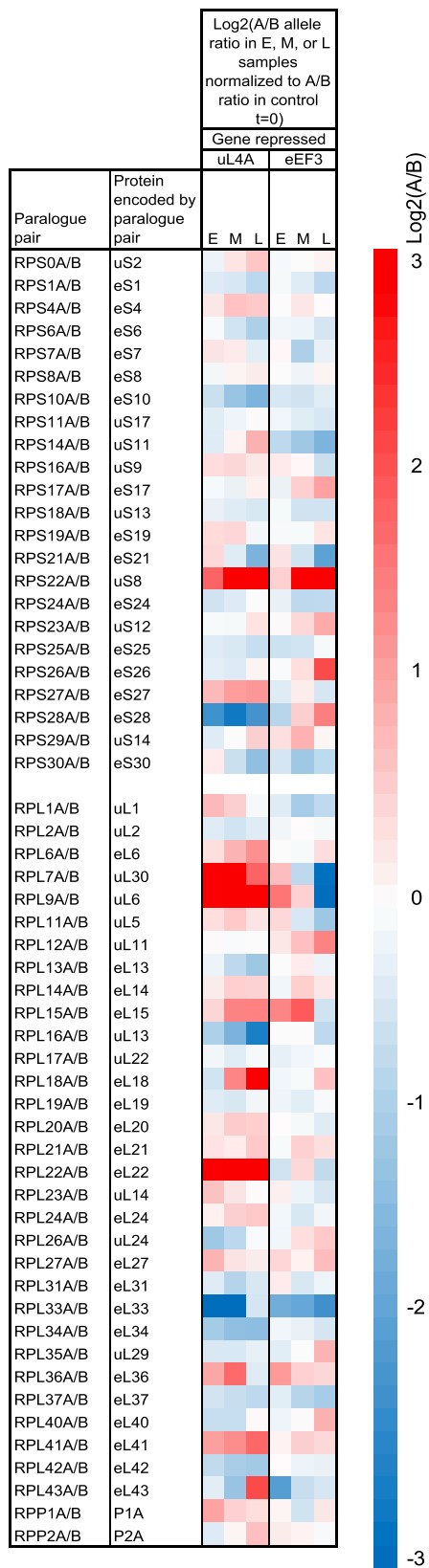

**FIG 6** Heatmaps for the relative abundances of mRNAs from the A and B paralogous alleles encoding the same r-proteins. The heatmap shows the ratio between the A and B alleles for a given r-protein normalized to the corresponding ratio in the control sample. Note that the scale of the heatmaps goes from −3 to +3.

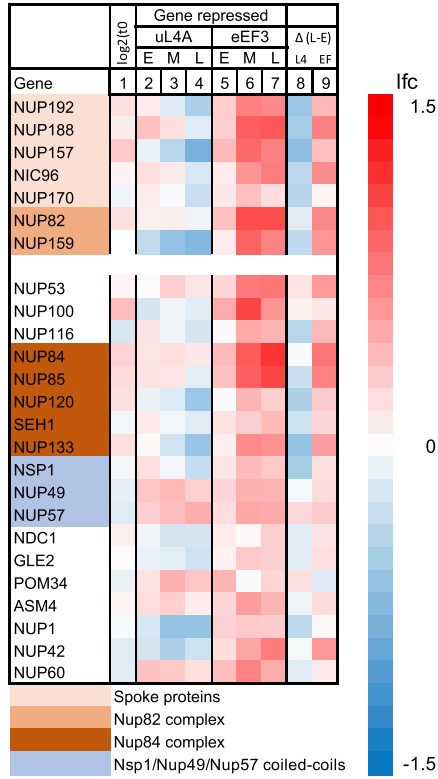

**FIG 7** Heatmaps for genes encoding nuclear pore proteins. The columns are organized as described in the legend to Fig. 3. Genes encoding components of subcomplexes of the nuclear pore are color-coded (60).

**Regulation of ribosomal protein mRNA abundance during stress.** Both the abundance and composition of r-protein mRNA are regulated differently by the two forms of stress. The abrogation of ribosome assembly by repressing the synthesis of uL4 prevents the assembly of the other newly synthesized proteins into stable, functional ribosomal particles (25). The disjunction of ribosome assembly initially decreased r-protein mRNA abundance (Fig. 4), presumably due to the accumulation of insoluble cytotoxic precipitates containing r-proteins and other proteins that directly or indirectly inhibited the transcription of r-protein genes (31). Interestingly, 2 h after the blocking of ribosome assembly, r-protein mRNA abundance increased (Fig. 4 and Fig. S2). We can think of two explanations. Since it is known that free r-proteins are attacked by proteasomes, we hypothesized that proteasome activity may have increased, thereby reducing the pool of free r-proteins and cytotoxic precipitates that repressed the r-protein synthesis during the initial period of nucleolar stress (32, 33). Second, the chaperone-mediated transport of nascent r-proteins to the nucleus may have increased, which would stabilize the r-protein mRNA (34).

The initial increase in r-protein mRNAs after stopping eEF3 synthesis was likely due to the nutritional shift up caused by transferring the culture to glucose medium. Nevertheless, 2 h after inducing translation stress, the r-protein mRNA abundance was repressed through the rest of the experiment. Perhaps as the number of eEF3 molecules fell below the number of ribosomes, the wait time for an eEF3 factor increased, temporarily stopping the movement of a fraction of the ribosomes and resulting in collisions between mobile and stationary ribosomes (Fig. 1A). This, in turn, may have triggered mRNA degradation (35, 36).

The difference in the r-protein transcriptome during the two types of stress correlated with the ratio between cell mass and ribosome number. During nucleolar stress, the ribosome formation was stopped, but the preexisting ribosomes kept synthesizing proteins, i.e., the cell mass-to-ribosome ratio increased. During translation stress, the ratio stayed constant because the synthesis of all proteins, including r-proteins, declined at the same rate as ribosome translocation was inhibited. Thus, the r-protein mRNA abundance was

upregulated only when the ratio between the cell mass and the ribosome number increased. It may also be important that the nucleolar stress is directed toward a nucle(o)lar process, while the translation stress is directed toward a cytoplasmic process.

**Differential expression of mRNA for r-protein paralogues.** More than two-thirds of the *S. cerevisiae* r-proteins are encoded by duplicated genes (A and B paralogous alleles) whose gene products can differ by a few amino acids. It is well established that the composition of paralogous r-proteins in individual ribosomes often varies in response to developmental or other external stimuli, thereby potentially generating "specialized ribosomes" with altered translational properties. However, a consensus on the biological significance of ribosome heterogeneity has not yet emerged (37–48).

To determine the effects of nucleolar and translation stress on the accumulation of mRNAs for paralogous r-proteins, we calculated if the ratio between the mRNA from the A and B paralogues changed during stress. Only minor changes in the paralogue ratio occurred for most r-proteins (Fig. 6). However, the relative amount of mRNA from paralogues encoding eS28, uL6, eL18, eL22, uL30, and eL33 changed by 2.5-fold or more during one or the other form of stress, while uS8 responded similarly to both stresses (Fig. 6). Since the ratio changed differently during the two types of stress for all proteins mentioned, except uS8, the change in paralogue compositions must have resulted from the imposition of stress, not the change in the growth medium. This is compatible with the notion that the relative expression of paralogous r-protein genes contributes to adapting the cell physiology during changing growth conditions.

**Genes encoding factors that support ribosome biogenesis and function are not coregulated with r-protein genes.** The ribosome assembly factors (Ribi) and translation factors are essential for the formation and function of ribosomes, respectively. Previous investigations suggested that the Ribi and r-protein genes are both parts of a large network regulated by the Sfp1 transcription factor (49, 50). However, our results showed that the genes for neither the Ribi nor the translation factors were coregulated with the r-protein mRNA during either ribosome or translation stress (Fig. 5), demonstrating that the coregulation of Ribi and ribosome auxiliary factors with r-proteins was not obligatory. We also noted that the expression of Ribi and r-protein genes was not coordinated during the inhibition of rRNA synthesis (31).

Ribosomal precursor particles are transported through the nuclear pores on their way to the cytoplasm, where final maturation into translation-competent ribosomes occurs. Hence, we analyzed the expression of mRNA for nuclear pore proteins. The mRNAs for these proteins were also not coregulated with the r-proteins transcriptome during nucleolar and translation stress (Fig. 7). We concluded that none of the genes that support ribosome biogenesis and function was coordinated with the r-protein mRNA abundance during interference with the functions they support.

**Other mRNAs of physiological importance.** Searching for other genes that are induced or repressed ($|lfc| \geq 1$) identified a diversity of GO terms (Data Set 4) that was in accordance with the established theorem that ribosome formation and function are central to the regulation of cell physiology.

**Implications.** Our data showed that the transcriptome evolves differently after inhibiting ribosome formation (nucleolar stress) and impeding ribosome translocation (translation stress), demonstrating that the response to a reduction of the protein synthesis capacity is not determined simply by the reduction of protein synthesis, but also by the process that lowers the translation capacity. Hence, we suggest that a given mutation simultaneously may affect both the synthesis of ribosomes and the ribosome function and that the combination of these two changes generates a unique transcriptome that gives each ribosomopathy mutation its unique phenotype.

## MATERIALS AND METHODS

**Strains and growth conditions.** Strains are listed in Table S1. Where indicated, the gene for r-protein uL4 or eEF3 is under the control of the *GAL1/10* promoter. We refer to these strains as Pgal-uL4 and Pgal-eEF3, respectively.

Cultures were grown asynchronously in YEPGal medium (1% yeast extract, 2% peptone, and 2% galactose) at 30°C. The optical density at 600 nm ($OD_{600}$) of a culture was measured using a 10-mm cuvette

in a Hitachi U1100 spectrophotometer (Hitachi High-Technologies Corp., Japan). When the galactose cultures reached an $OD_{600}$ of 0.8 to 1.0 (corresponding to $1.5 \times 10^7$ to $2 \times 10^7$ cells/mL), they were shifted to YPD medium (1% yeast extract, 2% peptone, and 2% glucose). Subsequently, cultures in either glucose or galactose medium were diluted as necessary with prewarmed medium to keep the $OD_{600}$ below 1.0 and prevent the cell density from reaching levels that would activate a transition to stationary phase. OD measurements made after the dilution(s) were multiplied with the dilution factors to generate a continuous growth curve. We did not expose the cultures to cycloheximide before harvest, because this treatment can influence both polysome content and mRNA abundance (51). All experiments were done in biological triplicates.

**RNA preparation and analysis.** Samples (1 OD unit) were taken from cultures before and at the indicated times after the shift to glucose medium (Fig. 2C). Total RNA was extracted using the Ribopure yeast kit (ThermoFisher, USA) following the manufacturer's protocol. rRNA integrity was checked with a Bio-Analyzer (Agilent Technologies, USA). One of the three Pgal-eEF3 12-h samples did not pass quality control and was not analyzed further. The other 23 RNA samples were enriched for mRNA using poly(A)-attached magnetic beads followed by paired-end RNA-seq library preparation using the TruSeq RNA sample prep kit (Illumina). One hundred nucleotides were sequenced using the HiSeq platform (Illumina) from both ends of each cDNA, and sequence reads were aligned against the *Saccharomyces cerevisiae* S288c reference genome using the Tophat2 aligner (52). Sequence annotation and read counts per gene were generated using HTseq (53) and Subread (54) and compared for consistency. We obtained a mean of $2.3 \times 10^7$ to $3.1 \times 10^7$ and $2.5 \times 10^7$ to $3.3 \times 10^7$ reads mapping to the *S. cerevisiae* BY4741 genome for each sample of Pgal-uL4 and Pgal-eEF3, respectively (Data Set 1). DESeq2 (55) was used for calculating differential gene expression between RNA samples relative to the 0-h sample. A gene was classified as a DEG if the FDR was <0.05 and the |lfc| was ≥1. The Python Sci-kit-learn was used for principal-component analysis. Heatmaps were generated by Excel software. The lfc and FDR of all genes with ≥50 reads at all times are summarized in Data Set 2A.

Gene ontology enrichment analysis was performed using the Gene Ontology Resource (http://geneontology.org/; release date 1 July 2022).

**Identification of genes expressed most and least differently during nucleolar and translation stress.** To identify the genes that were regulated most and least differently during nucleolar and translational stress, we ranked genes by fold change of expression for each time point and calculated the RVD by subtracting the gene rank values for individual genes in the Pgal-eEF3 samples from the rank values of the corresponding Pgal-uL4 samples. The absolute and actual RVD values for each gene were summed over three time points to calculate the absolute rank-sum and the rank-sum, respectively (Data Set 2B). Finally, the actual rank-sum was subtracted from the absolute value of the rank-sum to calculate the rank sign of each gene (Data Set 2C). Genes with the highest absolute rank-sum values differed the most in the fold change between the two strains (Pgal-uL4 and Pgal-eEF3) over the three time points of treatment. The genes with high rank-sign values were the genes that had a higher rank during repression of uL4 synthesis relative to repression of eEF3 synthesis at a specific time point but changed to a lower rank relative to eEF3 repression at other time points, and vice versa.

**Data availability.** The raw sequencing reads from this study have been submitted to the NCBI sequence read archive (SRA) under BioProject accession number PRJNA693823. The data are also available upon request.

## SUPPLEMENTAL MATERIAL

Supplemental material is available online only.
**DATA SET S1**, XLSX file, 0.8 MB.
**DATA SET S2**, XLSX file, 3 MB.
**DATA SET S3**, XLSX file, 0.1 MB.
**DATA SET S4**, XLSX file, 0.04 MB.
**FIG S1**, PDF file, 1.5 MB.
**FIG S2**, PDF file, 2.1 MB.
**FIG S3**, PDF file, 0.1 MB.
**TABLE S1**, PDF file, 0.03 MB.

## ACKNOWLEDGMENTS

This work was supported by grant 0920578 from the National Science Foundation to J.M.Z. and L.L. Additional funding was provided by an internal appropriation from the University of Maryland, Baltimore County, to L.L. We thank D. Etra for helpful discussions on data treatment and B. Traasdahl for help with the manuscript.

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
