## [Reviewer comments · mSystems]

Inhibiting ribosome assembly and ribosome translation have distinctly different effects on the abundance and paralogue composition of ribosomal protein mRNAs in *Saccharomyces cerevisiae*

Md Shamsuzzaman, Nusrat Rahman, Brian Gregory, Ananth Bommakanti, Janice Zengel, Vincent Bruno, and Lasse Lindahl

Corresponding Author(s): Lasse Lindahl, University of Maryland, Baltimore County

Review Timeline:

Submission Date:	November 9, 2022
Editorial Decision:	November 28, 2022
Revision Received:	December 9, 2022
Accepted:	December 15, 2022

Editor: Matthew Traxler

Reviewer(s): Disclosure of reviewer identity is with reference to reviewer comments included in decision letter(s). The following individuals involved in review of your submission have agreed to reveal their identity: Sunil Laxman (Reviewer #1)

Transaction Report:

DOI: <https://doi.org/10.1128/msystems.01098-22>

November 28, 2022

Dr. Lasse Lindahl
University of Maryland, Baltimore County
Department of Biological Sciences
1000 Hilltop Circle
Baltimore, Maryland MD 21250

Re: mSystems01098-22 (Inhibiting ribosome assembly and ribosome translation have distinctly different effects on the abundance and paralogue composition of ribosomal protein mRNAs in *Saccharomyces cerevisiae*)

Dear Dr. Lasse Lindahl:

Thank you for submitting your manuscript to mSystems. We have completed our review and I am pleased to inform you that, in principle, we expect to accept it for publication in mSystems. Both reviewers were positive about your study. However, I ask that you revise the manuscript and respond accordingly to the points raised by Reviewer 2.

Preparing Revision Guidelines

Sincerely,

Matthew Traxler

Editor, mSystems

Journals Department
Reviewer comments:

Reviewer #1 (Comments for the Author):

The authors have taken considerable efforts to resubmit a much-improved manuscript. The writing and description of the experiments are much clearer. The analysis is done more clearly, and most conclusions stated are appropriate. In general, the single point assessed, of what happens to overall ribosomal biogenesis if different aspects of translation (ribosome assembly vs translation initiation) are disrupted, is quite clearly answered. It is clear that the overall state of ribosome biogenesis is different, and this is demonstrated.

The last section of results (cell cycle arrest and "ribosome formation and function are central to the 929 regulation of cell growth and resource use.") only state the very obvious, and I would relegate that to two lines in the text, and a supplementary figure if required.

There is no further experimental analysis on what happens to overall translation fidelity or fitness as a consequence of these changes. That would have been ideal, to make stronger conclusions on what this might imply.

But in general, this is a nice story, and clarifies an important concept. The authors do need to proof this manuscript carefully).

Reviewer #2 (Comments for the Author):

Summary

In this revised manuscript, the authors clearly describe their use of a transcriptomic approach to determine whether cells respond similarly to nucleolar stress vs. translation stress generated by reduced expression of a ribosomal protein and a translation elongation factor, respectively. The clarity of the results is greatly improved as the authors leverage their data to (1) validate successful glucose-repression of the two targeted genes, (2) identify changes in mitochondrial gene expression likely caused by the carbon source switch, and (3) demonstrate that nucleolar stress and translation stress typically elicit opposite effects on expression of r-protein genes. Notably, some paralogues of these r-protein genes respond differently to the two stresses. Inclusion of improved heatmaps and deletion of other distracting figures results a streamlined narrative that is better supported by the highlighted data. In the long-run, these findings may inform our understanding of the diverse symptoms associated with various ribosomopathies. Since the eukaryotic microbe *S. cerevisiae* is used as the model organism for this study, and since the ribosomal machinery is highly conserved, the subject matter of this manuscript is likely to be of significant interest to mSystems readers.

Major Issues

The construction and use of growth curves could be clarified. In the results (lines 142-143), the authors state, "The sampling times were chosen to assure approximately equal growth rates of the two cultures at the time of sampling, estimated from respective growth curves" yet the methods section (lines 413-414) indicates "Cultures were diluted as necessary with pre-warmed media to keep the OD₆₀₀ <1.0." Which approach was used? Were the yeast cultures allowed to progress through the growth curve or not? Minor: In the figure 2 growth curves, the authors should double-check the y-axis labels, as the values suggest they may have graphed OD rather than logOD. For example, the data for the Pgal-eEF3 strain grown in glucose plateaus around log₂OD = 8, which would correspond to an actual OD value of 256!

It feels odd to dedicate a significant portion of the discussion to changes in expression of cell cycle and septum degradation genes when the corresponding figures are only available in the supplemental data. This part of the discussion could also be clearer. For example, in the first paragraph of cell cycle discussion (Page 12, Lines 358-369), the phrasing used is ambiguous regarding whether there was statistically significant enrichment of the GO term "septum digestion after cytokinesis". If so, the wording should be clarified, and the related heat map (S6A) from the supplemental data might be elevated and featured in the results section of the main paper. If not, this section of discussion might be cut from the paper, as the evidence does not sufficiently support it.

Minor Issues

There is no mention of Figure 7 in the results text. Since Figure 7 presents data on nuclear pore genes, it would make sense to reference that figure in the corresponding section of the results (Page 9, Lines 256-259). Also Page 12 line 354 contains a reference to Figure 6, but should it instead refer to Figure 7, which contains the nuclear pore gene expression data?

The discussion of possible explanations for changes in r-protein mRNA abundance could be clearer. For example, is one hypothesis that the cell would compensate for proteasome-mediated degradation of r-proteins by upregulating transcription of r-protein genes? And regarding ribosome collisions and mRNA degradation, it would be helpful to briefly clarify (or speculate on) how/why this would alter mRNA abundance of specific subsets of genes as opposed to resulting in a global reduction of total mRNA.

Page 12 line 371: Should delta lfc be greater than or equal to 1 as opposed to less than or equal to 1? If the authors did conduct GO term enrichment analysis on genes with delta lfc less than or equal to 1, this choice of cut off should be better justified.

Under Implications, the statement that "We suggest that a given mutation simultaneously affects both the synthesis of ribosomes and the ribosome function" (Page 13, Lines 397-398) could be rephrased to more clearly connect to and support adjacent sentences.

We thank the reviewers for their very helpful comments. We are pleased with the reviewers' positive reactions to the revised manuscript.

Reviewer #1 (Comments for the Author):

The authors have taken considerable efforts to resubmit a much-improved manuscript. The writing and description of the experiments are much clearer. The analysis is done more clearly, and most conclusions stated are appropriate. In general, the single point assessed, of what happens to overall ribosomal biogenesis if different aspects of translation(ribosome assembly vs translation initiation) are disrupted, is quite clearly answered. It is clear that the overall state of ribosome biogenesis is different, and this is demonstrated.

The last section of results (cell cycle arrest and "ribosome formation and function are central to the

929 regulation of cell growth and resource use.") only state the very obvious, and I would relegate that to two lines in the text, and a supplementary figure if required.

There is no further experimental analysis on what happens to overall translation fidelity or fitness as a consequence of these changes. That would have been ideal, to make stronger conclusions on what this might imply.

Both reviewers are critical of our discussion of the cell cycle. In general, we agree that this section does not live up to the general standard of the manuscript. Thus, we have eliminated the narrative section on "Cell Cycle" and Table S6.

We also condensed the narrative of "Other mRNAs", but do not think that it should be eliminated completely, since it completes the analysis.

But in general, this is a nice story, and clarifies an important concept. The authors do need to proof this manuscript carefully).

Reviewer #2 (Comments for the Author):

Summary

In this revised manuscript, the authors clearly describe their use of a transcriptomic approach to determine whether cells respond similarly to nucleolar stress vs. translation stress generated by reduced expression of a ribosomal protein and a translation elongation factor, respectively. The clarity of the results is greatly improved as the authors leverage their data to (1) validate successful glucose-repression of the two targeted genes, (2) identify changes in mitochondrial gene expression likely caused by the carbon source switch, and (3) demonstrate that nucleolar stress and translation stress typically elicit opposite effects on expression of r-protein genes. Notably, some paralogues of these r-protein genes respond differently to the two stresses. Inclusion of improved heatmaps and deletion of other distracting figures results a streamlined narrative that is better supported by the highlighted data. In the long-run, these findings may inform our understanding of the diverse symptoms associated with various ribosomopathies. Since the eukaryotic microbe *S. cerevisiae* is used as the model organism for this study, and since the ribosomal machinery is highly conserved, the subject matter of this manuscript is likely to be of significant interest to mSystems readers.

Major Issues

The construction and use of growth curves could be clarified. In the results (lines 142-143), the authors state, "The sampling times were chosen to assure approximately equal growth rates of the two cultures at the time of sampling, estimated from respective growth curves" yet the methods section (lines 413-414) indicates "Cultures were diluted as necessary with pre-warmed media to keep the OD600<1.0." Which approach was used? Were the yeast cultures allowed to progress through the growth curve or not? Minor: In the figure 2 growth curves, the authors should double-check the y-axis labels, as the values suggest they may have graphed OD rather than logOD. For example, the data for the Pgal-eEF3 strain grown in glucose plateaus around $\log_2 OD = 8$, which would correspond to an actual OD value of 256!

We have clarified the recording of the growth curve data and that the culture dilutions are necessary to keep the culture from entering stationary phase which would prevent the identification of the specific stress-induced changes to the transcriptome.

It feels odd to dedicate a significant portion of the discussion to changes in expression of cell cycle and septum degradation genes when the corresponding figures are only available in the supplemental data. This part of the discussion could also be clearer. For example, in the first paragraph of cell cycle discussion (Page 12, Lines 358-369), the phrasing used is ambiguous regarding whether there was statistically significant enrichment of the GO term "septum digestion after cytokinesis". If so, the wording should be clarified, and the related heat map (S6A) from the supplemental data might be elevated and featured in the results section of the main paper. If not, this section of discussion might be cut from the paper, as the evidence does not sufficiently support it.

We have eliminated the discussion of the cell cycle; see our response to Reviewer 1's comments.

Minor Issues

There is no mention of Figure 7 in the results text. Since Figure 7 presents data on nuclear pore genes, it would make sense to reference that figure in the corresponding section of the results (Page 9, Lines 256-259). Also Page 12 line354 contains a reference to Figure 6, but should it instead refer to Figure 7, which contains the nuclear pore gene expression data?

We apologize for the error. It has been corrected and the Figure numbering corrected to follow the narrative.

The discussion of possible explanations for changes in r-protein mRNA abundance could be clearer. For example, is one hypothesis that the cell would compensate for proteasome-mediated degradation of r-proteins by upregulating transcription of r-protein genes?

We have added specificity to this hypothesis.

And regarding ribosome collisions and mRNA degradation, it would be helpful to briefly clarify (or speculate on) how/why this would alter mRNA abundance of specific subsets of genes as opposed to resulting in a global reduction of total mRNA.

Page 12 line 371: Should delta lfc be greater than or equal to 1 as opposed to less than or equal to 1? If the authors did conduct GO term enrichment analysis on genes with delta lfc less than or equal to 1, this choice of cut off should be better justified.

This section has been rewritten; see above.

Under Implications, the statement that "We suggest that a given mutation simultaneously affects both the synthesis of ribosomes and the ribosome function" (Page 13, Lines 397-398) could be rephrased to more clearly connect to and support adjacent sentences.

We have modified the text.

December 15, 2022

Dr. Lasse Lindahl
University of Maryland, Baltimore County
Department of Biological Sciences
1000 Hilltop Circle
Baltimore, Maryland MD 21250

Re: mSystems01098-22R1 (Inhibiting ribosome assembly and ribosome translation have distinctly different effects on the abundance and paralogue composition of ribosomal protein mRNAs in *Saccharomyces cerevisiae*)

Dear Dr. Lasse Lindahl:

Your manuscript has been accepted, and I am forwarding it to the ASM Journals Department for publication. For your reference, ASM Journals' address is given below. Before it can be scheduled for publication, your manuscript will be checked by the mSystems production staff to make sure that all elements meet the technical requirements for publication. They will contact you if anything needs to be revised before copyediting and production can begin. Otherwise, you will be notified when your proofs are ready to be viewed.

Publication Fees:

If you would like to submit a potential Featured Image, please email a file and a short legend to mSystems@asmusa.org. Please note that we can only consider images that (i) the authors created or own and (ii) have not been previously published. By submitting, you agree that the image can be used under the same terms as the published article. File requirements: square dimensions (4" x 4"), 300 dpi resolution, RGB colorspace, TIF file format.

We recognize that the video files can become quite large, and so to avoid quality loss ASM suggests sending the video file via <https://www.wetransfer.com/>. When you have a final version of the video and the still ready to share, please send it to mSystems staff at mSystems@asmusa.org.

Sincerely,

Matthew Traxler
Editor, mSystems

Journals Department
Supplemental Material Fig S1: Accept
Supplemental Material Dataset 3: Accept
Supplemental Material Table S1: Accept
Supplemental Material Dataset 4: Accept
Supplemental Material Dataset 2: Accept
Supplemental Material Dataset 1: Accept
Supplemental Material Fig S2: Accept
Supplemental Material Figure S3: Accept